# Demonstration of 12.5 Mslot/s 32-PPM Underwater Wireless Optical Communication System with 0.34 Photons/Bit Receiver Sensitivity

Xiaotian Han [1,2], Peng Li [1], Guangying Li [1], Chang Chang [1], Shuaiwei Jia [1,2], Zhuang Xie [1], Peixuan Liao [1,2], Wenchao Nie [1] and Xiaoping Xie [1,2,*]

1   State Key Laboratory of Transient Optics and Photonics, Xi'an Institute of Optics and Precision Mechanics, Chinese Academy of Sciences, Xi'an 710119, China; hanxiaotian@opt.ac.cn (X.H.); niewenchao@opt.ac.cn (W.N.)
2   School of Future Technology, University of Chinese Academy of Sciences, Beijing 101408, China
*   Correspondence: xxp@opt.ac.cn

**Abstract:** High-capacity, long-distance underwater wireless optical communication (UWOC) technology is an important component in building fast, flexible underwater sensing networks. Underwater communication with light as a carrier has a large communication capacity, but channel loss induced by light attenuation and scattering largely limits the underwater wireless optical communication distance. To improve the communication distance, a low-power 450 nm blue continuous wave (CW) laser diode (LD)-based UWOC system was proposed and experimentally demonstrated. A communication link was designed and constructed with a BER of $3.6 \times 10^{-3}$ in a total link loss of 80.72 dB in $c = 0.51 \text{ m}^{-1}$ water with a scintillation index (S.I.) equal to 0.02 by combining with 32-pulse-position modulation (32-PPM) at a bandwidth of 12.5 MHz and single photon counting reception techniques. The allowable underwater communication distance in Jerlov II ($c = 0.528 \text{ m}^{-1}$) water was estimated to be 35.64 m. The attenuation lengths were 18.82, which were equal at link distances of 855.36 m in Jerlov I ($c = 0.022 \text{ m}^{-1}$) water. A receiving sensitivity of 0.34 photons/bit was achieved. To our knowledge, this is the lowest receiving sensitivity ever reported under 0.1 dB of signal-to-noise ratio (SNR) in the field of UWOC.

**Keywords:** underwater optical wireless communication; high sensitivity; photon counting; single photon detection; bule-green laser





## 1. Introduction

The ocean, covering approximately 71% of the earth's surface, is rich in biological and mineral resources. In recent years, the increasing scarcity of terrestrial resources has led to an increasing focus on the exploration of the oceans. For this reason, countries around the world have established ocean observation platforms and developed high-performance underwater observation equipment to gradually enhance the understanding and study of the oceans. With the rapid development of this oceanographic equipment, large amounts of oceanographic data need to be transmitted back to land, and high-performance underwater equipment needs to be able communicate with each other over a long distance [1].

Underwater communication can be divided into wired and wireless communication according to the different transmission media [2]. Wired communication uses cables or fiber-optic cables to connect communication nodes for data transmission, which greatly limits the mobility of the observation equipment. Wireless communication uses seawater as a transmission medium to transmit signals. As seawater is a good conductor, the attenuation of an electromagnetic wave is so severe that traditional radio communication cannot be adapted to the underwater environment [3]. Unlike radio communication, hydroacoustic communication with less energy attenuation has been widely used for underwater wireless signal transmission. However, the low rate and the large time delays greatly limit the

effectiveness and timeliness of hydroacoustic communication [4]. In addition, acoustic carriers are not suitable for use for air–water cross-media communication scenarios [5].

Underwater wireless optical communication (UWOC) systems, with high bandwidth, lightweight, and low power consumption, have an advantage over hydroacoustic communication in solving the problem of air–water high-speed transmission [6]. Blue-green light is usually used for underwater optical communication due to the relatively low light attenuation in seawater [7]. For UOWC, effects including seawater absorption, scattering, and turbulence make the optical signal attenuate severely during transmission, which limits the communication distance to two hundred meters [5,8]. In Table 1, we summarize the schemes and performance of published underwater wireless optical communication systems. The optical communication system equipped with a high-energy laser and high-sensitivity photon-counting model achieves the lowest received optical power and the highest total link loss [9]. However, limited by the repetition frequency of the pulsed laser, the communication rate of this optical communication system is only 12 kbps. In addition to the reception sensitivity of the communication system, the signal-to-noise ratio related to channel loss and environmental noise, detector noise, and detection schemes is another important factor. Due to the high link loss, after the optical signal is transmitted over a long distance in the underwater channel, the signal intensity drops to the level of a photon comparable to the receiver noise, which will cause the bit error rate to be too high to establish a reliable underwater optical communication link. Due to its outstanding noise suppression ability, the scheme of coherent detection has been widely used in free optical communication [10–12]. However, the fast decoherence of light in water limits the use of coherent optical communication for long-distance underwater communication. A potential way to improve the system's sensitivity against low SNR and high underwater channel loss is to directly make use of single photon detection and photon counting [13–15].

**Table 1.** Comparison of the published UWOC system.

| Light Source | Power | Modulation | Detectors | Rate | Received Optical Power | SNR | Received Photons per Bit | References |
|---|---|---|---|---|---|---|---|---|
| 517 nm LD | 100 mW | OOK | PCM | 1.30 Mbps | −84.1 dBm | +20 dB | ~1.0 | [16] |
| 520 nm LD | 19.4 mW | OOK | PD | 2.7 Gbps | −8.24 dBm | +1.8 dB | ~$10^5$ | [17] |
| 532 nm SSL | 1.5 W | 256-PPM | PCM | ~12 kbps | −105.04 dBm | +2.5 dB | ~0.33 | [9] |
| 450 nm LD | 174 μW | PPM | MPPC | 5 MHz | −39.19 dBm | +2.1 dB | ~$10^4$ | [18] |
| 520 nm LD | 7.3 mW | OOK | PD | 500 Mbps | −19.77 dBm | +2.6 dB | ~$10^5$ | [19] |
| 520 nm LD | 10 mW | OOK | SiPM | 1 Gbps | −40.9 dBm | - | ~$10^2$ | [14] |
| 532 nm LD | 10 mW | BPSK | SiPM | 500 Mbps | −48.2 dBm | - | ~$10^2$ | [20] |
| 450 nm LD | 0.47 mW | 32-PPM | PCM | 1.9 Mbps | −84.0 dBm | +0.1 dB | 0.34 | This work |

LD: laser diode; SSL: high-power solid-state laser; OOK: on–off-keying; PPM: pulse position modulation; PD: photodiode; MPPC: muli-pixel photon counter; SiPM: Si Photomultiplier; PCM: photon-counting model.

Here, we designed and demonstrated an underwater high-sensitivity wireless optical communication system combined with PPM and single photon counting techniques. The ultimate performance of the optical communication system was Evaluated by combining experimental results and Monte Carlo analysis tools. The following sections discuss the details of the theoretical analysis, experimental setup, and testing results.

## 2. System Characteristics

### 2.1. Absorption and Scattering

Light absorption and scattering, inherent optical transmission properties in the underwater channel, would cause the light energy decrease exponentially when the light beam moves in the water. Light scattering would also cause the optical beam to spread in time and space [21]. Two wavelength-related feature parameters, $a(\lambda_0)$ and $b(\lambda_0)$ (named

the absorption coefficient and scattering coefficient, respectively), are used to describe the absorption and scattering effects [22]. The attenuation coefficient $c(\lambda_0)$ can then be given as:

$$c(\lambda_0) = a(\lambda_0) + b(\lambda_0) \tag{1}$$

where $\lambda_0$ is the light wavelength. We usually evaluate the performance of UWOC systems under a specific wavelength. The optical characteristics parameters of various water types are shown in Table 2.

The reason for this is that light scattering involves photon interactions with water molecules, particulate matter, and other dissolved substances in the water [23]. When a photon collides with a particle, the direction of motion will deviate; this deviation angle is called the scattering angle, and is expressed as $\varphi$. The scatterers in ocean water are randomly distributed, and so the volume scattering phase function, $\beta(\varphi, \lambda_0)$, is usually used to describe the angle distribution of the scattering and to predict the moving direction of scattered photons [24]. Moreover, we obtain an expression $b(\lambda)$ by integrating $\varphi$ over all angles [21,25,26]:

$$b(\lambda_0) = 2\pi \int_0^\pi \beta(\varphi, \lambda_0) \sin \varphi d\varphi \tag{2}$$

By normalizing the volume scattering phase function $\beta(\varphi, \lambda_0)$ with $b(\lambda_0)$, we then arrive at the scattering phase function:

$$\widetilde{\beta}(\lambda_0, \varphi) = \frac{\beta(\lambda_0, \varphi)}{b(\lambda_0)} \tag{3}$$

Many different types of analytical phase functions have been used over the years. The Henyey–Greenstein (HG) function has been widely used in the research of light scattering in an underwater channel over the years [27,28]. In more recent years, the Fournier–Forand (FF) scattering phase function [29–32] has been used, and has been shown to be in good agreement with actual water measurements. The FF scattering phase function is an approximate analytic form of the phase function of an ensemble of particles that have a hyperbolic particle size distribution, with each particle scattering according to the anomalous diffraction approximation to exact Mie theory. The phase is beneficial for modeling both the forward scattering and backscattering of natural waters.

$$\widetilde{\beta}_{FF}(\varphi) = \frac{1}{4\pi(1-\delta)^2\delta^v} \left[ v(1-\delta) - (1-\delta^v) + [\delta(1-\delta^v) - v(1-\delta)] \sin^{-2}\left(\frac{\varphi}{2}\right) \right]$$
$$+ \frac{1-\delta_{180}^v}{16\pi(\delta_{180}-1)\delta_{180}^v} \left( 3\cos^2 \psi - 1 \right) \tag{4}$$

where

$$v = \frac{3 - \mu}{2} \tag{5}$$

$$\delta = \frac{4}{3(n-1)^2} \sin^2\left(\frac{\varphi}{2}\right) \tag{6}$$

Here, $n$ is the real index of refraction of the particles, $\mu$ is the slope parameter of the hyperbolic distribution, and $\delta_{180}$ is $\delta$ evaluated at $\varphi = 180$ deg.

$$B_p = 1 - \frac{1 - \delta_{90}^{v+1} - 0.5\left(1 - \delta_{90}^v\right)}{(1 - \delta_{90})\delta_{90}^v} \tag{7}$$

where $\delta_{90}$ is $\delta$ evaluated at $\varphi = 90$ deg. The backscatter fraction, $\beta_p$, which represents the polar angle of a photon scattered greater than $90°$, is very helpful for confirming the parameters $n$ and $\mu$ in the actual water.

**Table 2.** Optical characteristics parameters for the various water types [33].

| Water Types | $\lambda_0$ [nm] | $a$ [m$^{-1}$] | $b$ [m$^{-1}$] | $c$ [m$^{-1}$] | $L_{ext}$ [m] | Albedo $\omega_0$ |
|:---:|:---:|:---:|:---:|:---:|:---:|:---:|
| Jerlov I | 450 | 0.018 | 0.0038 | 0.022 | 45.87 | 0.17 |
| Jerlov IA | 450 | 0.0221 | 0.00631 | 0.028 | 35.71 | 0.23 |
| Jerlov IB | 450 | 0.0235 | 0.068 | 0.092 | 10.87 | 0.74 |
| Jerlov IC | 450 | 0.105 | 0.514 | 0.619 | 1.62 | 0.83 |
| Jerlov II | 450 | 0.0241 | 0.504 | 0.528 | 1.89 | 0.95 |
| Jerlov III | 450 | 0.0388 | 2.38 | 2.419 | 0.41 | 0.98 |

Parameters: $L_{ext}$: extinction length; $\omega_0 = b/c$.

### 2.2. Underwater Turbulence

In the underwater channel, in addition to light absorption and light scattering, the turbulence induced by the temperature gradients, salinity gradients, air bubbles, and ocean mixing processes will result in scintillation and phase change of the received optical signal, leading to an increase in the bit error rate (BER). Since the intensity-modulated and direct detection (IM/DD) communication system was used in this experiment, we only considered the optical scintillation effect caused by the underwater channel. In terms of the effect of scintillation, the intensity of turbulence is usually quantified by the scintillation index (S.I.) or the probability density distribution (PDF) of received optical power [34]. A large number of underwater turbulence experiments have proved that the log-normal probability density function $f(P)$ can well represent the PDF of received optical power from weak to strong turbulence under the aperture averaging effect [34–36].

$$f(P) = \frac{1}{P\sigma\sqrt{2\pi}} \exp\left(-\frac{\left(\ln(P_r/P_0) + \sigma^2/2\right)^2}{2\sigma^2}\right) \tag{8}$$

where $P_0$ is the mean received optical power, and $\sigma^2$ is the S.I. defined by:

$$\sigma^2 = \frac{\langle P_r^2 \rangle - \langle P_r^2 \rangle^2}{\langle P_r^2 \rangle^2} \tag{9}$$

Here, < > is the mean operator.

### 2.3. The Theoretical Analysis of M-PPM Photon-Counting Communication System

M-PPM photon-counting receiver has been shown to achieve high communication sensitivity [9,37]. In the M-PPM symbol, a single pulse in one of the M transmission slots represents $\log_2(M)$ bits of information. When the PPM system works, the laser can operate at both high peak power and low average power, with the number of signal photons received corresponding to the energy of the emitted single laser pulse energy [38]. For the photon-counting receiver, assuming the signal optical power incident on a single-photon detector is $P_r$, the average photon arrival rate will be [39]:

$$\lambda_s = \frac{\gamma P_r}{h\nu} \tag{10}$$

where $\gamma$ is the quantum efficiency of a single-photon detector, $h$ is Planck's constant, and $\nu$ is the frequency photon. Considering the influence of background optical power and the single-photon detector's dark count, the average noise photon rate is given as:

$$\lambda_n = \frac{\gamma P_b}{h\nu} + \lambda_d \tag{11}$$

where $P_b$ is the background optical power, and $\lambda_d$ is the dark count rate. Then, the average number of counted photons in a period of time $t = T_s$ can be arrived at:

$$\rho = \lambda T_s \tag{12}$$

Since the single-photon detector is in a silent state during the dead time and cannot respond sequentially to arriving photons, the maximum number of photons detected by the single-photon detector in a period of t is $\rho_{max} = \lfloor T_s / \tau + 1 \rfloor$, with $\lfloor x \rfloor$ denoting the largest integer that is smaller than $x$ [40]. The photocount distribution of an active quenching single-photon detector with a dead time of $\tau$ during the time interval of $T_s$ is given by [41]:

$$f(\rho, \lambda_\rho) = \begin{cases} \sum\limits_{i=0}^{\rho} \psi(i, \lambda_{\rho+1}) - \sum\limits_{i=0}^{\rho-1} \psi(i, \lambda_\rho) & \rho < \rho_{max} \\ 0 & \rho \geq \rho_{max} \end{cases} \tag{13}$$

where

$$\psi(i, \lambda_\rho) = \frac{\lambda^i (t - \rho\tau)^i}{i!} e^{-\lambda(t - \rho\tau)} \tag{14}$$

In particular, the mean and variance of the photocount distribution are [40]:

$$\mu_K = (\rho_{max} - 1) - \sum\limits_{\rho=0}^{\rho_{max}-2} \sum\limits_{i=0}^{\rho} \psi(i, \lambda_{\rho+1}) \tag{15}$$

$$\sigma_K^2 = \sum\limits_{\rho=0}^{\rho_{max}-2} \sum\limits_{i=0}^{\rho} (2\rho_{max} - 2\rho - 3)\psi(i, \lambda_{\rho+1}) - \left( \sum\limits_{\rho=0}^{\rho_{max}-2} \sum\limits_{i=0}^{\rho} \psi(i, \lambda_{\rho+1}) \right)^2 \tag{16}$$

The symbol error probability (SER) of the M-PPM system is therefore [42]:

$$P_s = \frac{1}{M} \sum\limits_{\rho=0}^{\rho_{max}} L(\rho) \left[ M f_{Y|X}(\rho|0) - F_{Y|X}(\rho|0)^M + F_{Y|X}(\rho - 1|0)^M \right] \tag{17}$$

Combined with Equation (14), where

$$f_{Y|X}(\rho|1) = f\left(\rho, \lambda_\rho^s + \lambda_\rho^n\right) \tag{18}$$

$$f_{Y|X}(\rho|0) = f\left(\rho, \lambda_\rho^n\right) \tag{19}$$

$$F_{Y|X}(\rho|0) = \sum\limits_{i=0}^{\rho} f_{Y|X}(\rho|0) \tag{20}$$

$$L(\rho) = \frac{f_{Y|X}(\rho|1)}{f_{Y|X}(\rho|0)} \tag{21}$$

Once the M-PPM symbol is detected, the symbol information will be demapped into $\log_2(M)$ binary bits, and the resulting bit error ratio (BER) is [43]:

$$P_b = \frac{M}{2(M-1)} P_s$$
$$= \frac{1}{2(M-1)} \sum\limits_{\rho=0}^{\rho_{max}} L(\rho) \left[ M f\left(\rho, \lambda_\rho^s\right) - \left[ \sum\limits_{i=0}^{\rho} f\left(\rho, \lambda_\rho^n\right) \right]^M + \left[ \sum\limits_{i=0}^{\rho} f\left(\rho - 1, \lambda_{\rho-1}^n\right) \right]^M \right] \tag{22}$$

Based on the theoretical model of the photon-counting communication system with PPM, we designed a 32-PPM underwater wireless optical communication system and carried out experimental verification.

## 3. Experimental Setup

The proposed experimental configuration of the underwater photon-counting communication system is illustrated in Figure 1. The system is combined with a directly modulated laser diode (LD) transmitter, simulated underwater channel, and photon-counting receiver. The transmitter included LD, an attenuator, and collimating optics. The computer maps pseudorandom binary sequence (PRBS) to 32-PPM symbols. The 32-PPM symbols from the computer were fed into the arbitrary waveform generator (AWG) to be converted into a modulated analog signal that was amplified by an RF amplifier (SHF 100 AP) by 10 dB. In the experiment, we set the clock frequency and level amplitude of the AWG to 12.5 MHz and 400 mW, respectively. Then, the modulator (Thorlabs LDM9LP) drove the amplified electrical signal onto the LD to generate the modulated optical signal. The bias tee superimposed a bias current that was controlled by a constant voltage source. The transmitted Gaussian beam generated from continuous wave LD (CW laser, Thorlabs LP450-SF15), after being fed into a subsequent variable attenuator (Thorlabs V450A), was expanded by a beam collimator system (NA = 0.25, $f$ = 10.9 mm). The output of the collimator was launched into a 5 m water tank. The mode field diameter launched was 2 mm, which means that the initial light divergence angle was 286 μrad. At the received end, an enclosure housing the receiver optics was used to shade the background light. The receiver enclosure's input aperture diameter was 3 cm. A 4f magnification system based on L1 and L2 was used to reshape the beam. For limiting the field-of-view and rejecting the background light, an iris had a variable in the L1 focal plane that could control the angular field-of-view between ~3 mrad and 158.5 mrad to reduce the impact of ambient noise and the multipath effect. Subsequently, the 1:1 beam splitter split the light into two paths. One of the paths was focused on the optical power measurement device, another path was focused on the photomultiplier tube (PMT, Hamamatsu H10682-210), and a high-resolution oscilloscope (OS) captured the output signal from PMT. After synchronization, the captured signal was demodulated and decoded into binary bits. The BER was then calculated by comparing the decoded bits to transmitted bits. The 10 nm optical filter (Thorlabs FBH520-10) in the PMT path was used to reject the out-of-band background.

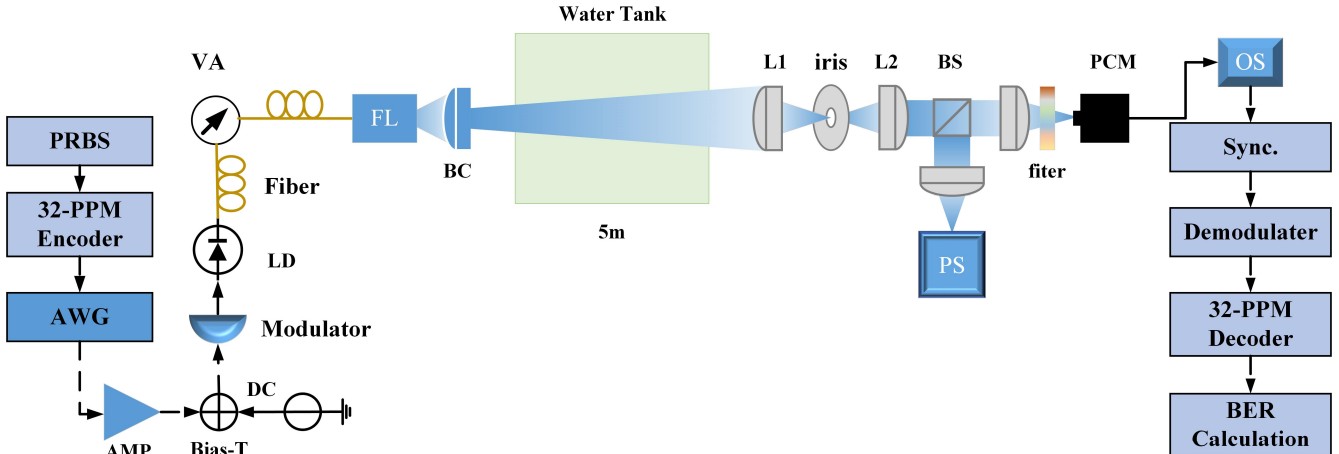

**Figure 1.** Schematic of the experimental setup. PRBS: pseudo-random bit sequence; AWG: arbitrary waveform generator; AMP: RF amplifier; LD: laser diode; VA: variable attenuator; FL: fiber launch; BC: beam collimator; BS: beam splitter; PCM: photon-counting module; PMD: power measurement device; OS: oscilloscope.

### 3.1. LD and Transmitter

#### 3.1.1. LD Light Source

Figure 1 shows the view of LD used in this experiment and the performance results tested. As shown in Figure 2a, this setup selected the directly modulated single-mode fiber-pigtailed laser. Figure 2b shows that the central wavelength of a single LD is 450.1 nm

with a full width at half maximum (FWHM) of 2 nm. The tested 3 dB bandwidth of the LD was about 1 GHz, as shown in Figure 2c. Figure 2d shows the tested results of V-I and P-I curves when the operating temperature of LD was 25 °C. The threshold current was 27 mA and the mean output optical power was 15.0 mW when the bias current was set at 66 mA.

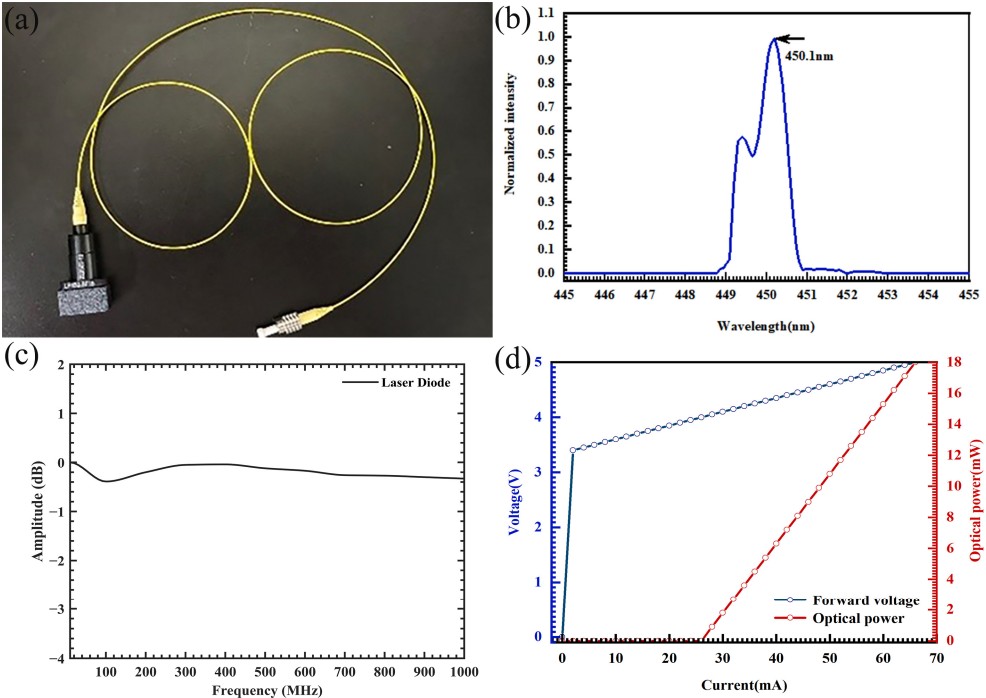

**Figure 2.** Transmitter blue LD: (**a**) single mode fiber-pigtailed LD, (**b**) the optical spectral property, (**c**) 3 dB bandwidth of LD, (**d**) VI and P–I curves when the operating temperature was 25 °C.

### 3.1.2. Transmitted Waveform

The data rate of the communication system is limited by the modulation bandwidth. Several studies have demonstrated that the M-PPM can achieve high-sensitivity communication by increasing the size of modulation order M, but the requirement for modulation bandwidth will be higher [38]. The higher modulation bandwidth means a shorter slot duration time, which leads to fewer received photons in a signal slot. To trade off the modulation bandwidth and M, in this current system, we chose 32-PPM for our modulation scheme. The transmitted data were organized into frames. The frame structure is shown in Figure 3. To achieve the slots and symbol synchronization, a special frame header was fixed at the beginning of the frame. The receiver can distinguish the slot sequences and the start of a frame from the received data sequence by identifying the frame header. The following 500 symbols are a payload sequence.

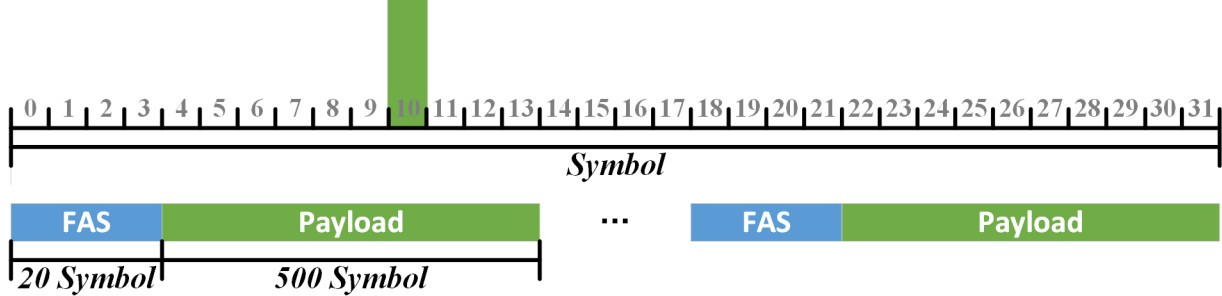

**Figure 3.** The signal frame structure.

A frame acquisition sequence (FAS) with 24 PPM symbols (0, 3, 1, 2, 1, 3, 2, 0, 0, 3, 2, 1, 0, 2, 1, 3, 1, 0, 3, 2, 3, 2, 1, 0) for 4-PPM are provided by the standard for Optical Communications Coding and Synchronization recommended by the Consultative Committee for Space Data System [44]. Later on, the FAS is modified by reference [13] into [0, 3, 1, 2, 1, 3, 2, 0, 0, 3, 2, 1, 0, 2, 1, 3, 1, 0, 3, 2, 3, 2, 1, 0, 0, 1, 2, 3, 2, 3, 0, 1, 3, 1, 2, 0, 1, 2, 3, 0, 0, 2, 3, 1, 2, 1, 3, 0]. We converted 48 4-PPM symbols into 32-PPM symbols as shown in Figure 4 using the zero-fill operation as the FAS in this experimental setup. In this study, the modulation bandwidth in the system was set at 12.5 MHz, which means the signal slot duration time was 80 ns. After loading the waveform signal, the measured average transmitting optical power was 0.47 mW, and the single-pulse energy was 1.2 nJ.

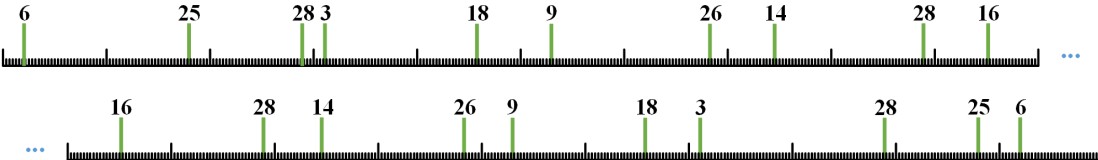

**Figure 4.** Frame acquisition sequence.

### 3.2. Underwater Channel Characterization

In the experiment, the water tank with dimensions of $5 \times 0.8 \times 0.8$ m shown in Figure 5a was used to simulate an underwater channel, and Figure 5b was the view of received light spots with an emitted spot diameter of 2 mm after transporting in the tank water. In particular, Magnesium hydroxide ($Mg(OH)_2$) powder, as shown in Figure 5c, was added to the water to vary the attenuation coefficient. The effect of turbulence was enhanced by the wave-making air pump shown in Figure 5d that was used to generate the air bubbles. Here, we did not consider the effect of turbulence caused by temperature and salinity gradients.

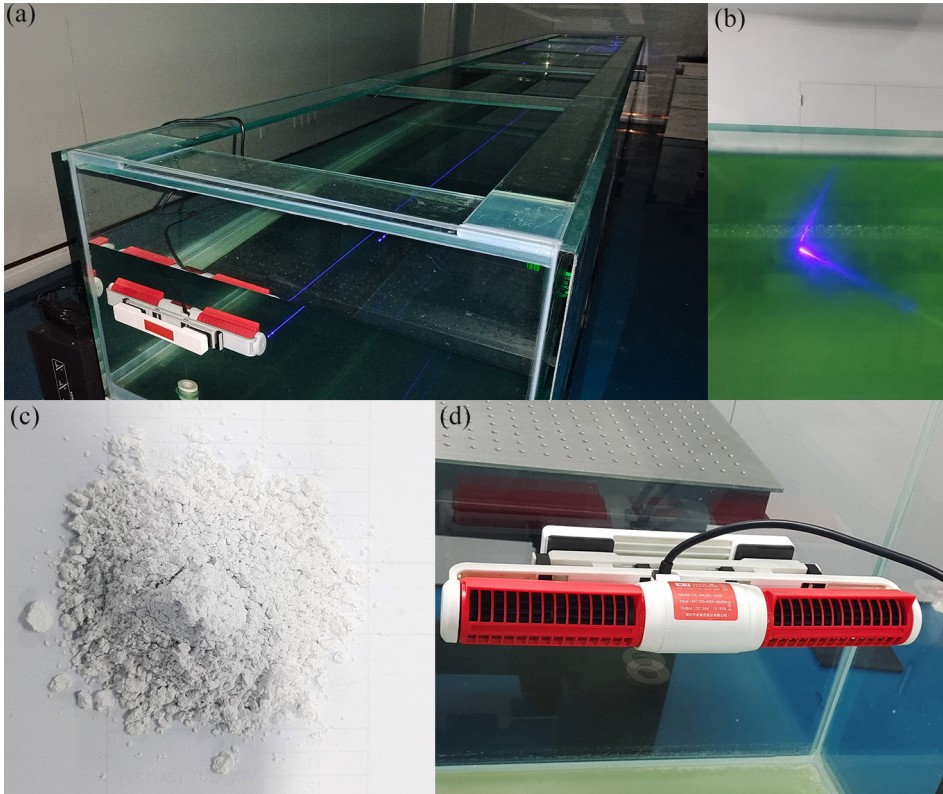

**Figure 5.** The simulated underwater channel: (**a**) the water tank, (**b**) the view of the received optical spot in the water, (**c**) $Mg(OH)_2$ powder, (**d**) the wave-making air pump.

The transparency of water can greatly affect the performance of UWOC, thus the effect of power loss in the underwater channel is one of the important factors needed to be considered due to the variability of the underwater channel. Parameters characterizing the quality of water include the absorption coefficient $c$, the scattering coefficient $b$, the scattering albedo $\omega$, and the attenuation coefficient $c$. In this experiment, we applied Jerlov's water-typed parameters, and the value of $a$, $b$, and $c$, $\omega$ are shown in Table 1. The Beer–Lambert law is commonly used to describe the power loss as a function of transmission distance in UWOC [22], which can be:

$$P(z) = P(0)e^{-c(\lambda_0)z} \tag{23}$$

where $z$ is the communication length. To obtain the attenuation coefficient $c$ of the experimental water tank. As shown in Figure 6a, we tested the power loss of measured water over different transmission distances. The blue prismatic points are the measured data, and the red fitting curve is based on the Beer–Lambert model with $c = 0.51$ m$^{-1}$, here we ignored the geometric loss because the received optical spot was smaller than the receiver aperture size in this experiment. The water quality in the experimental tank was close to that of the Jerlov II water according to the data in Table 1. However, the Beer–Lambert law assumes that all scattered light is lost from the propagation beam and no multiply scattered light returns to the beam, which results in the gain because the photons scattering back into the central beam are not considered [45]. Instead, in consideration of energy conservation, the radiance transmission equation (RTE), which is derived to theoretically describe all photons moving through the water along a path toward a given direction [46], is given as:

$$\cos\theta' \frac{dI(z,\lambda,\varphi',\theta')}{dz} = -c(\lambda)I(z,\lambda,\varphi',\theta')$$
$$+ \int_0^\pi \int_0^{2\pi} \beta(z,\lambda,\varphi \to \varphi', \theta \to \theta')I(z,\lambda,\varphi',\theta')\sin\psi_1 d\theta' d\varphi' \tag{24}$$

where $I(z, \lambda, \varphi, \theta)$ is the radiance energy in the direction with the scattering angle $\varphi_i$ ($i = 1,2$)and the azimuth angle $\theta_i$ ($i = 1,2$), and the angle $\varphi_i$ is obtained by the scattering phase function described in Equation (4). In Equation (24), the first term on the right is Beer's law loss, and the second term is the gain from light scattering from the angles $\varphi$, $\theta$ into $\varphi'$, $\theta'$. The analytical solution of the equation is extremely difficult since it involves integrals and derivatives. The Monte Carlo (MC) simulation tool has been widely used to solve the RTE [24,46]. In this approach, the interaction of each photon with the medium is statistically modeled, and the photon propagation paths from the transmitter to the receiver are traced step by step. In this study, we traced $2.7 \times 10^9$ photons (each photon has an energy of $4.42 \times 10^{-19}$ J, thus, the total energy of the traced photons is 1.2 nJ) from the light source to a receiver with an MC simulator based on the Jerlov II water parameters shown in Table 1 and the parameters described in in the experimental setup of transmitter and receiver. The water quality parameters include the attenuation coefficient $c$ and the albedo $\omega$. The parameters of the transmitter include light wavelength, initial divergence angle, and laser spot radius. The parameters of the receiver include field-of-view (FOV) and aperture diameter. The simulation results are presented in Figure 6b, which is the curve of power loss and 3 dB underwater channel bandwidths as a function of transmission distance. The simulation results show that link loss was −85.73 dB when the transmission distance was 37.88 m.

In UOWC, the communication bandwidth limits the maximum communication rate. The communication bandwidth is determined by the channel bandwidth, the transmitter, and the receiver bandwidth. We measured the transmitter bandwidth at the LD and transmitter, which was about 1 GHz. For underwater channels, the 3 dB bandwidth is determined by the multipath effect generated by scattering [46,47]. On the one hand, for a given transmitter, receiver, and water quality parameters, the multipath effect increases with the transmission distance due to the increased scattering events. On the other hand, an appropriate optical system for the transmitter and receiver can limit the multipath effect, and a reduced light divergence angle, FOV, and receiving aperture size can reduce the

influence of the multipath effect and increase the 3 dB bandwidth of the communication system [48]. In this study, we simulated the 3 dB channel bandwidth under Jerlov II-type water by setting the initial divergence angle as 286 μrad, the FOV as 100 mrad, and the aperture diameter as 3 cm. The simulation results shown in Figure 6b suggest that the 3 dB bandwidth of the tested beam was less than 384.62 MHz at a transmission distance of less than 37.88 m.

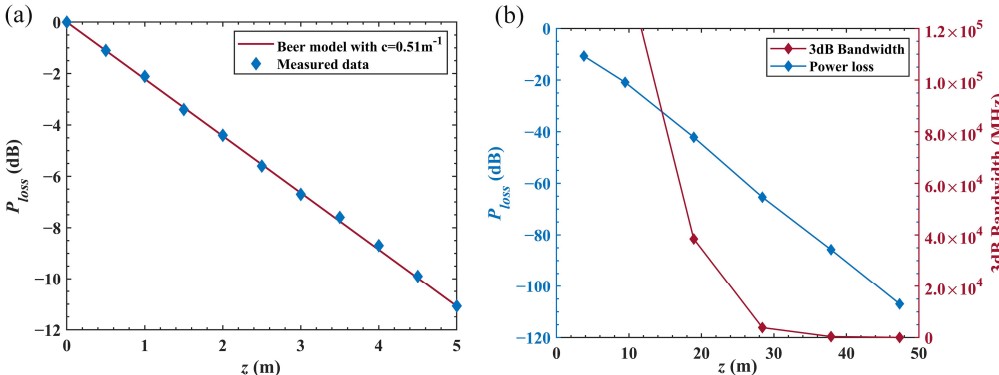

**Figure 6.** The results of the test and simulation: (**a**) the measured transmittance of the tested water, (**b**) Normalized power loss, and 3 dB underwater channel bandwidths for Jerlov II.

In this part, we tested the air-bubble-induced turbulence in the experimental water tank. As mentioned above, the interaction between the laser beam and turbulent medium causes amplitude variations (scintillation) in the optical beam carrying the information, which results in fading of the received optical power. Based on the log-normal turbulence model described by Equation (7), as shown in Figure 7a, we measured the received optical power over 2 min (red curve) and calculated the normalized received optical power fluctuation (blue curve), and the scintillation index (S.I.) was calculated. Figure 7b shows the fit of the log-normal with the measured data histogram.

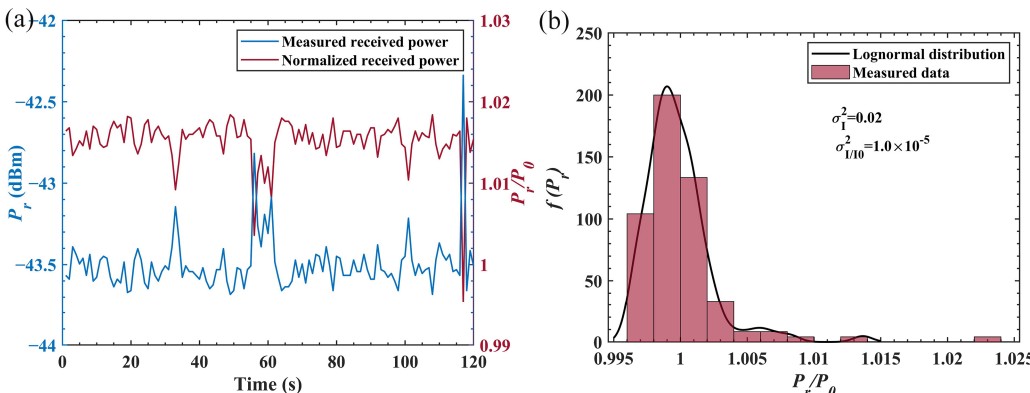

**Figure 7.** The test results of turbulence: (**a**) the measured and normalized power variation as a function of time, (**b**) the histogram of the measured data with the PDF of normalized power and scintillation.

### 3.3. Photon Counting with PCM

The PCM includes an analog photomultiplier (PMT) and a discriminator. Due to the photocathode being sensitive to a single photon, a PCM can be used for single-photon detection and photon counting [49]. In PCM, the PMT is used to convert the incident photons to high-amplitude electric pulses, and the discriminator set at some threshold level is used to analyze the PMT output. Once the electrical output exceeds the set threshold, a detection occurs. By setting an appropriate threshold, most spikes of current consistent with the amplification of thermal electrons that escaped from the dynodes of PMT (especially those dynodes that are closer to the anode) may be filtered out. One important issue is that the electrical current at the output of the PCM is no longer a continuous high voltage

output when a photon is incident on the photocathode, and the output electrical pulse has a finite width and a fixed amplitude. In this experiment, an off-the-shelf photon-counting module (HAMAMATSU, H10682, $40 \times 22 \times 36$ mm) was selected as the device. Figure 8 demonstrates the internal structure of the device. The diameter of the effective area is 8 mm. As shown in Figure 8c, the yellow curve is the output waveform of the conventional analog photodetector, and the green curve shows the output waveform of the PCM. Since the operating characteristics of the PCM are dependent on wavelength, we tested the performance parameters including the output pulse width, the dead time, the dark, and the distribution of photocounts of the PCM used in this experiment at 450.1 nm.

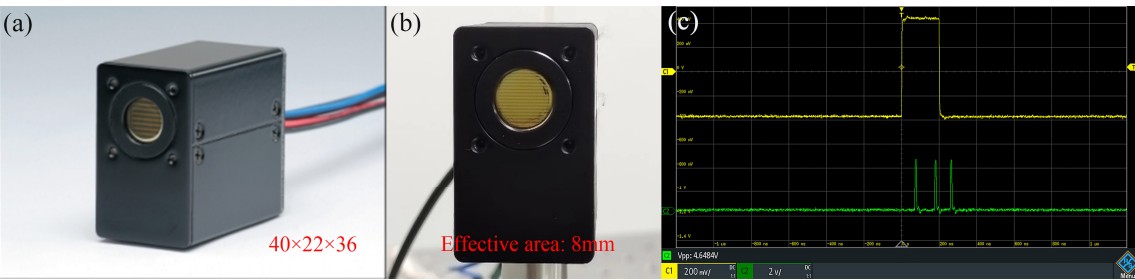

**Figure 8.** The appearance of the photon-counting module: (**a**) dimensions of the module, (**b**) the front view of the module, (**c**) the output waveform from the analog output of conventional photodetector (yellow curve) and PCM (green curve).

### 3.3.1. Output Pulse Width and Dead Time

The output pulse width and dead time of the PCM are two important parameters which directly influence the bandwidth of the photon-counting receiver. For a PCM, if a single-photon detection event occurs, the detection system will become blind to the incoming photon. The time after an initial detection during which no new input photons can be detected is named dead time. A typical dead time for a PCM ranges from microseconds to nanoseconds. Short dead times are significant for the high-rate photon-counting receiver. As shown in Figure 9a, we sampled the analog output of PCM using high-speed oscilloscopes under high input power. In addition, they are influenced by afterpulses caused by the elastic scattering of electrons on the first dynode of PMT and the ionization of residual gas molecules in the PMT tube, which leads to the output pulse spreading. Moreover, due to the effect of the detector itself and pulse-counting electronics, the dead time can randomly change. Therefore, we calculated the mean value of the output pulse width and dead times. They were 11.7 ns and 18.5 ns, respectively.

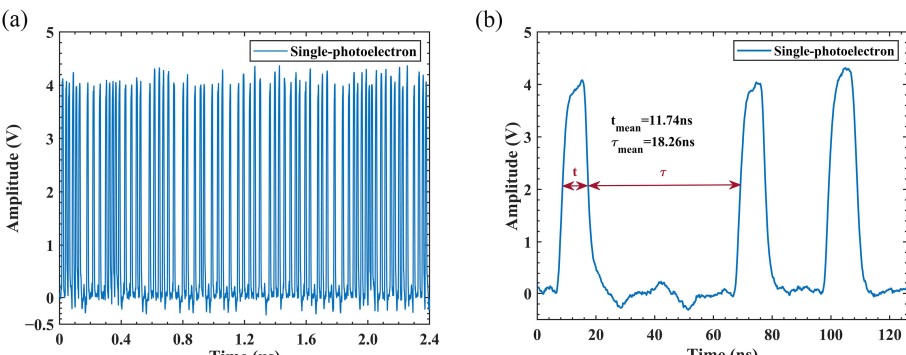

**Figure 9.** The characteristics of PMT output pulse: (**a**) the single-photoelectron waveform over 2.4 μs, (**b**) the single-photoelectron waveform over 120 ns.

### 3.3.2. Background and Dark Counts

The output pulse and dead time, background and dark count can also not be ignored. The dark count is the average number of counts registered by a detector per second when

all input light to the detector is blocked. In practical applications, the dark count and background count cannot be clearly distinguished because the background photons cannot be adequately blocked from reaching a detector. We counted the number of output pulses under the actual experimental environment. The counting time was set to 80 ns which corresponded to the slot time of the communication system. As statistical results are shown in Figure 10, the background and dark counts are 0.001 photo counts per slot time and 0.007 per symbol (32-PPM) time, respectively.

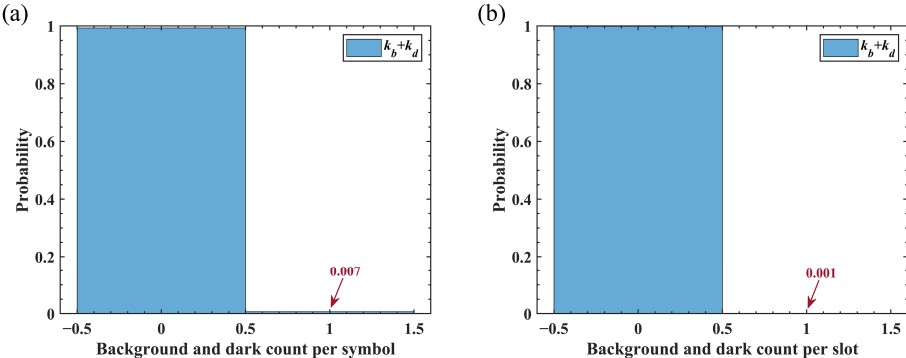

**Figure 10.** The measured background and dark count: (**a**) the histogram of background and dark counts per plot time (80 ns), (**b**) the histogram of background and dark counts per symbol time (2.56 μs).

### 3.3.3. Distribution of Photocounts

According to Formulas (15) and (16), we measured the mean value $\mu_k$ and the variance $\sigma_{2K}$ of output photocounts for the PCM as functions of average photon arrival rate λs when the photocount time was 3 μs. As shown in Figure 11a, it is obvious that the photo count distribution of a PCM cannot be modeled as an ideal Poisson process whose mean and variance are always equal, which is mainly due to the influence of dead time. The mean value of photocounts gradually converges to 100 as $\lambda_s$ increases, that is, the total number of counted photons during a counted interval $[0, T_s]$ cannot exceed kmax = $[T_s/\tau] + 1$. The measured and simulated $\mu_k$, $\sigma_{2K}$ have an obvious difference, and the difference becomes more significant as $\lambda_s$ increases. At low photon arrival rate $\lambda_s$, the measured and simulated $\mu_k$ and $\sigma_K{}^2$ are approximately equal, but after the $\lambda_s$ reaches approximately $2 \times 10^7$ photocounts/s, the difference between the measured value and simulated value is obvious. The probability density function (PDF) obtained in Equation (13) is plotted in Figure 11b and compared with experimental results for different values of the photocount time interval. In this figure, time intervals $T_s = 3$ μs, 1.5 μs, 0.3 μs are set, and $\lambda_s = 2 \times 10^7$ photoelectrons/s. As shown in Figure 11b, the experimental and simulation results have a good agreement.

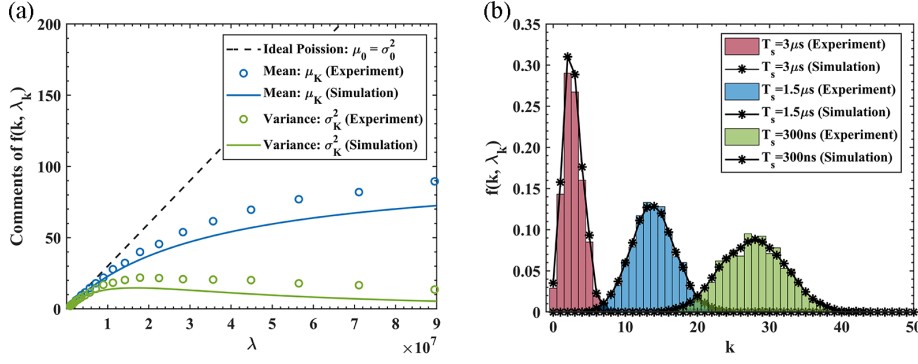

**Figure 11.** PCM photocounts characteristics: (**a**) Mean and variance of photocounts with $t = 3$ μs, (**b**) probability density distribution of PCM photocounts for $T_s = 3$ μs,1.5 μs, 0.3 μs, $\lambda_s = 2 \times 10^7$ photocounts/s.

## 4. Results of the Water Tank Experiment

We investigated the communication performance of the proposed underwater photon-counting communication system. In this communication system, the 32-PPM modulation was used, and the slot frequency of the system was set as 12.5 MHz. The received optical power was attenuated from −76 dBm to −86 dBm by improving the input voltage of the variable attenuator. Figure 11 only shows the output signal by oscilloscope that for sampling rate was 250 MS/s at received optical was −76 Bm and −84 dBm, respectively. By comparing Figure 12a,c, we intuitively find that the average number of photons at the signal time slots is higher at the received optical power of −76 dBm than at −84 dBm, but the maximum photon counts per slot do not exceed four photons. Moreover, the photon counts at the empty time slots increase as received optical power increases.

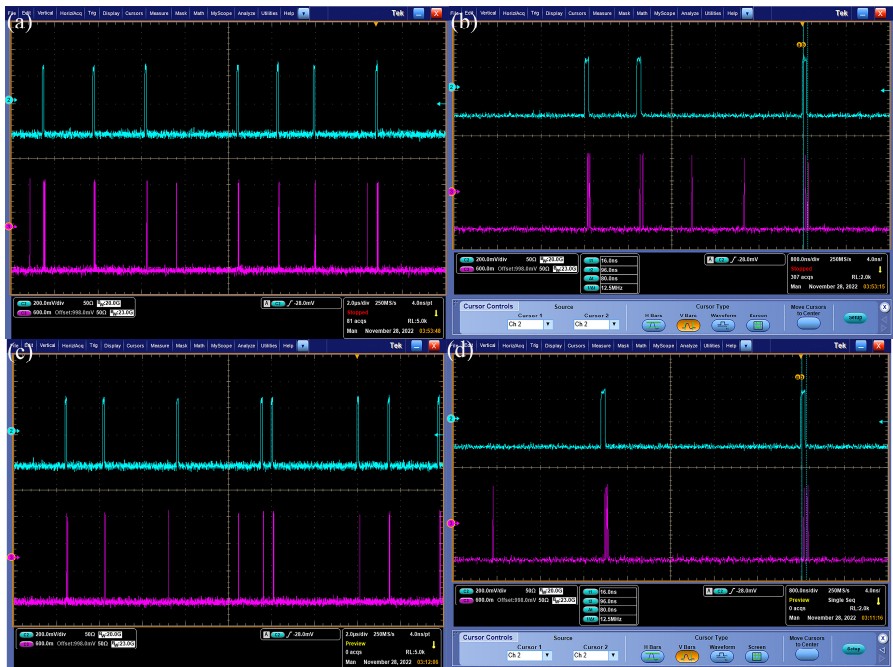

**Figure 12.** The signal sampled by the oscilloscope at the sampling rate was 250 MS/s: the display resolution of the oscilloscope is 2.0 μs/div (**a**) and the display resolution of the oscilloscope is 800.0 ns/div (**b**) at a received optical power of −76 dBm; the display resolution of the oscilloscope is 2.0 μs/div (**c**) and the display resolution of the oscilloscope is 800.0 ns/div (**d**) at a received optical power of −84 dBm.

To further analyze the performance of the photon-counting communication system proposed in this study, the frequency histogram of photocounts per symbol times under the differently received optical powers was calculated. As shown in Figure 13, the horizontal coordinate of the picture indicates the number of counted photons per the signal slot of 32-PPM symbols, and the vertical coordinate indicates the probability of the number of counted photons of each symbol. The frequency of photon counts at the signal slots is represented by the light red histogram, the frequency of photon counts at the empty time slots is represented by the blue histogram, and the dark red histogram shows the overlapping part. As shown in the frequency histogram of $k_s$, for 12.5 Mslot/s, the probability that the number of counted photons is equal to 3 or 4 increases with the received optical power. However, the number of counted photons never exceeds 4. The mean photon counts in signal slots, $k_s$, and the mean number of counted photons in the empty slots, $k_n$ are labeled in Figure 13.

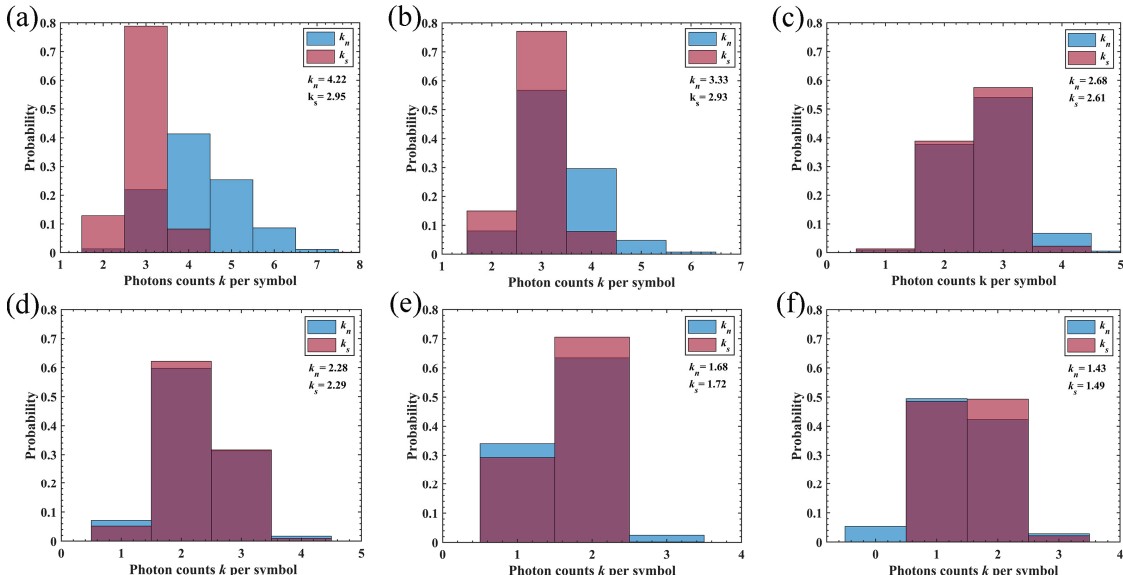

**Figure 13.** The frequency histogram of photocounts per symbol times for 32-PPM photon-counting communication system vs. received optical power: (**a**) −86 dBm, (**b**) −84 dBm, (**c**) −82 dBm, (**d**) −80 dBm, (**e**) −78 dBm, (**f**) −76 dBm.

Figure 14 shows that for 12.5 Mslot/s, $k_s$, $k_b$, and the signal-to-noise ratio (SNR) are a function of received optical power. The measured kb increases with the received optical power. However, because the PCM's saturation is dependent on the dead time, the measured $k_s$ increases with the received optical power with nearly constant proportionality below −78 dBm. In this case, the SNR of the receiver decreases with received light power until it falls to a negative value above −82 dBm.

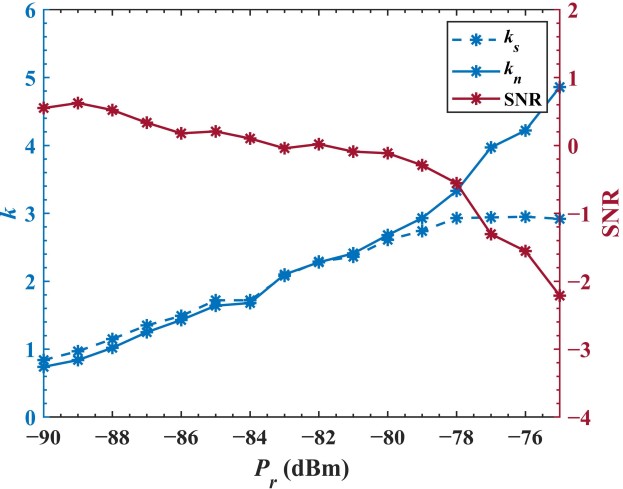

**Figure 14.** The mean number of photons, $k_s$, $k_n$, and SNR as a function of received optical power.

Figure 15 shows the measured maximum likelihood decision BER as a function of optical power $P_r$ and the number of counted photons $k_s$. As shown in Figure 15a, the corresponding BER is $3.6 \times 10^{-3}$ when the received optical power is −84 dBm, which is below the FEC limit (BER of $3.8 \times 10^{-3}$). At this received optical power, as shown in Figure 14, the mean number of counted photons in a signal slot is 1.72 photons where SNR is 0.1 dB. Therefore, we can conclude that the tested photon-counting-based UWOC system has achieved 80.72 dB link loss with S.I. equal to 0.02 under a communication rate of about 1.9 Mbps.

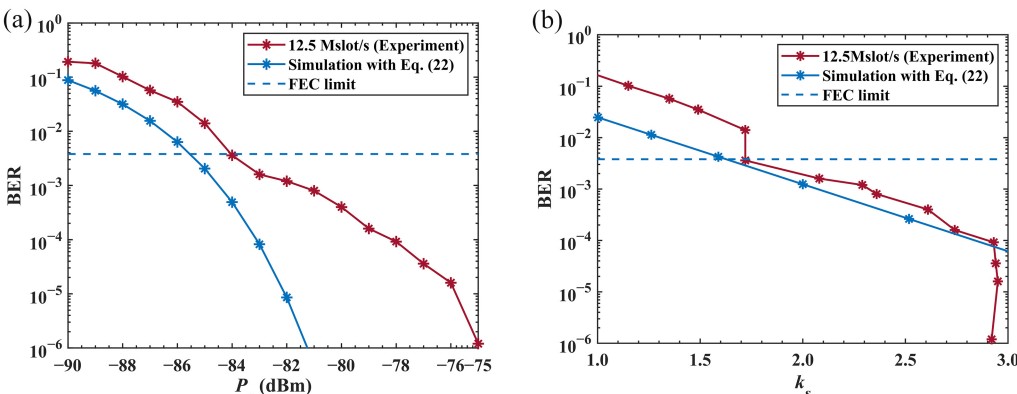

**Figure 15.** BER as functions of (**a**) received optical power $P_r$ and (**b**) photon counts $k_s$.

## 5. Conclusions

In this paper, we analyzed the performance of the 32-PPM photon-counting communication system under specific water conditions and designed an experimental setup combining the blue CW laser modulated by 32-PPM and a photon-counting module with an average dead time of 18.26 ns. The attenuation coefficient in this experiment was $c = 0.51$ m$^{-1}$, which was close to that of the Jerlov II water ($c = 0.528$ m$^{-1}$). The total link loss can reach 80.72 dB. The communication link for a 35.64 m distance was achieved based on the Monte Carlo simulation results shown in Figure 6b. The attenuation length (AL = $c \times z$) was 18.82 and the equivalent link distances are 855.36 m (Jerlov I water) and 672.14 m (Jerlov IB water). To reduce the influence of the multipath effect and background light, we limited the initial light divergence angle and field-of-view (FOV) to 286 μrad and 100 mrad, respectively. The experimental results revealed that the photon-counting communication system can achieve a communication rate of 1.9 Mbps at a signal-to-noise ratio of 0.1 dB. The mean number of experimentally received photon counts in a signal slot was 1.72, with one signal pulse representing 5 bits of information for a 32-PPM system. Therefore, the photon-counting receiver described in this paper can achieve a receiving sensitivity of 0.34 photons/bit. To the best of our knowledge, this is the lowest receiving sensitivity ever reported at a communication rate of 1.9 Mbps and 0.1 dB of SNR. It should be noted that the optical quantum efficiency of the photon-counting module used in this paper was 10%, and if the quantum efficiency of the detector could be increased, this photon-counting communication system would achieve greater link loss.

In future work, signal processing technology to reject background noise will be investigated. In addition, the highly accurate system of underwater narrow laser pointing and tracking will also be the focus of research.

**Author Contributions:** X.X. contributed to the conception and organized the manuscript of the study; X.H. and P.L. (Peng Li) performed the experiment; G.L. and C.C. contributed significantly to analysis and manuscript preparation; S.J. and Z.X. helped perform the analysis with constructive discussions. P.L. (Peixuan Liao) and W.N. revised the manuscript. All authors have read and agreed to the published version of the manuscript.

**Funding:** This research was funded by National Key R&D Program, grant number 2021YFC2800504 and National Key R&D Program, grant number 2022YFC2806000.

**Institutional Review Board Statement:** The study did not require ethical approval.

**Informed Consent Statement:** The study did not involve humans.

**Data Availability Statement:** Not applicable.

**Acknowledgments:** We are grateful to the four anonymous reviewers for their constructive and helpful comments on the manuscript.

**Conflicts of Interest:** No conflict of interest exists in the submission of this manuscript, and the manuscript is approved by all authors for publication. I would like to declare on behalf of my

co-authors that the work described was original research that has not been published previously, and not under consideration for publication elsewhere, in whole or in part. All authors listed have approved the manuscript that is enclosed.

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
