# Peer review of "Demonstration of 12.5 Mslot/s 32-PPM Underwater Wireless Optical Communication System with 0.34 Photons/Bit Receiver Sensitivity"

_photonics, doi:10.3390/photonics10040451_

Round 1

Reviewer 1 Report

This paper have demonstrate a UOWC system with 0.34photons/bit receiver sensitivity. This work can be valuable for the implementation of long distance UOWC system. But there are still several problems that should be clarified before its publication.

1. In Sec. 2.1 and 2.2, the authors introduced the absorption, scattering, and turbulence charateristics of sea water. But these thereotical presentation cannot be related to the experimental setting. For example, the authors introduced the FF scattering phase function in Eq. (4). How this function is related to their experimental setting? On the other hand, in Eq. (8), they introduced the log-normal PDF to model the oceanic turbulence. In the referece, this PDF model is used to charaterise the thermal and salinity fluctuation induced turbulence. But in their experiment, they use it to describe the air bubble induced turbulence. The authors should explain why they did these settings. 

2. Table 1, SPAD is not used in table 1.

3. In line 147, the mathematical description of "not larger than" should be modified to the usual form.

4. The results of the BER equation Eq. (22) should be compared with the experimental data.

5. Fig.1, FC in Fig. 1 should be modified to FL.

6. In Fig.6 (a), the experimental data coincide with the Beer-Lamber law. Why did the authors do the MC simulation in Fig. 6(b)? The authors want to demontrate the experimental data is identical with Beer-Lambert or MC? They should explain it clearly. 

Author Response

Dear Reviewer:

  Thank you for your letter and your comments concerning our manuscript. We have studied the comments carefully and have made correction which we hope meet with approval. Revised portions are marked in red on the attached paper.

  1. In Sec. 2.1 and 2.2, the authors introduced the absorption, scattering, and turbulence charateristics of sea water. But these thereotical presentation cannot be related to the experimental setting. For example, the authors introduced the FF scattering phase function in Eq. (4). How this function is related to their experimental setting? On the other hand, in Eq. (8), they introduced the log-normal PDF to model the oceanic turbulence. In the reference, this PDF model is used to charaterise the thermal and salinity fluctuation induced turbulence. But in their experiment, they use it to describe the air bubble induced turbulence. The authors should explain why they did these settings. 

Response: In this paper, we used a tank which is a length of 5m to simulate the absorption, scattering and turbulence. In addition, the attenuation coefficient was varied by the addition of magnesium hydroxide powder (Mg(OH)2). These are described in detail in section 3.2 of this paper.

In Section 3.2 of the paper, we obtained the attenuation coefficient of the water in the tank by fitting the experimental data with Beer-Lambert's law. In order to obtain the transmission characteristics of the experimental system over longer distances under the same water quality condition, we used Metro-Carlo (MC) tool to simulate the characteristics of power attenuation and bandwidth response of the wireless optical communication system designed in this paper after transmitting 50m in water with an attenuation coefficient of 0.51m-1. The FF scattering phase function is used in MC simulations to obtain the probability distribution of the scattering angle after the photon has been scattered.

In an underwater channel, turbulence causes the optical fluctuation effect which leads to fluctuation in the light energy reaching the receiver. The fluctuations in the optical power of the signal are usually described using a log-normal probability density function. In the actual seawater environment, the communication distance of an underwater wireless optical communication system is within 200 meters, currently. Turbulence effects due to temperature and salinity fluctuations can be ignored, which is mentioned in the 2020 paper entitled " Channel model and performance analysis of long-range deep sea wireless photon-counting communication" by HUANG J et al.

  1. Table 1, SPAD is not used in table 1.

Response: This was a misspelling, and we have removed it in this paper.

  1. In line 147, the mathematical description of "not larger than" should be modified to the usual form.

Response: The mathematical description in line 154 has been modified to ‘ρmaxTs/τ+1û, with ëxû’.

  1. The results of the BER equation Eq. (22) should be compared with the experimental data.

Response: The results of the BER equation Eq. (22) are compared with the experimental data in Figure 15.

  1. Fig.1, FC in Fig. 1 should be modified to FL.

Response: FC in Fig. 1 has been modified to FL.

  1. In Fig.6 (a), the experimental data coincide with the Beer-Lamber law. Why did the authors do the MC simulation in Fig. 6(b)? The authors want to demonstrate the experimental data is identical with Beer-Lambert or MC? They should explain it clearly. 

Response: At short distances, the attenuation characteristics of optical power in the water medium are consistent with the Beer-Lamber law, whereas, at long transmission distances, the attenuation characteristics of optical power in water are not consistent with the Beer-Lamber law, as mentioned in reference [45]. To obtain the attenuation coefficient of water in the tank in the experiment, we measured the received optical power at different transmission distances and fitted it using the Beer-Lamber law to obtain the attenuation coefficient of water in the tank. In this paper, according to the experimental results, the overall link loss of this experimental system can reach 80.72dB, limited by the experimental conditions, the optical power attenuation after the 5m water tank is only -11 dB. Therefore, to evaluate the maximum communication distance of an underwater wireless optical communication system designed in this paper at the same water quality condition, we used the MC to simulate the characteristics of power attenuation and bandwidth response of the wireless optical communication system, the simulation results is shown in Fig. 6(b).

Reviewer 2 Report

Please check the attachment

Author Response

Dear Reviewer:

  Thank you for your letter and your comments concerning our manuscript. We have studied comments carefully and have made correction which we hope meet with approval. Revised portions are marked in blue on the attached paper.

  1. Some abbreviations should be defined when writing them for the first time such as, but not limited to, CW in the abstract.

Response: The full name of CW has been added in line 17 and line 178

  1. The authors mention that: “Blue-green light is the “optical windows’ for underwater optical communication due to the low light attenuation in seawater”; maybe this statement is weak as red light as shown potential performance also in underwater communications. So, that statement should be corrected.

Response: The statement has been corrected “Blue-green light is usually used for underwater optical communication due to the relatively low light attenuation in seawater” in line 53.

  1. (4) needs to be clearly explained. Also, some parameters in Eq. (11) need to be defined too.

Response: We have added a note to Eq. (4).

The parameter η has been γ Eq. (11), which is the quantum efficiency.

  1. What are the parameters used at AWG such as the Vpp, and sampling frequency?

Response: The parameters used at AWG have been added in line 174.

  1. The used modulation order throughout the experimental test is 32-PPM, what is the reason behind that selection?

Response: The relationship between the time slot frequency B of the PPM and the data transmission rate Rs is given:

The parameters M and L, which represent bite solution and modulation order, are equal to 5 and 32 respectively in this paper. To ensure that the data transmission rate of the system is greater than 1Mbps, and to ensure the highest communication sensitivity, we set the modulation order to 32.

  1. In conclusion, the authors mentioned that M-PPM while only 32-PPM was selected throughout the paper; that should be corrected.

Response: This problem has been corrected in line 449.

Reviewer 3 Report

This paper provides experimental results of underwater optical communication using a blue laser with 450 nm wavelength, and demonstrates the possibility of high sensitivity detection.

The following should be considered during the preparation of the next draft.

1. There are other published papers that report the use of 450 nm blue laser and a gigabits per second data rates. For example, the 2017 paper entitled "blue laser diode enables underwater communication at 12.4 Gbps" by T-C Wu et al. provides experimental result to demonstrate > 10 m transmission. It is not clear if these papers are not cited because they do not provide path loss and sensitivity figures.

2. The connection between the detailed analysis included in this paper and the experimental results is not clear. The text should clearly show how the analysis is used or verified.

3. Discussions of results could be improved, with proper justifications and conclusions. A marked-up manuscript is attached that points out a few of these locations.

4. The paper should be properly edited to improve consistency and reduce grammatical errors. The attached manuscript highlights many places where obvious grammatical corrections are warranted.

Blue Laser Diode Enables Underwater Communication at 12.4 Gbps

Author Response

Dear Reviewer:

  Thank you for your letter and your comments concerning our manuscript. We have studied comments carefully and have made correction which we hope meet with approval. Revised portions are marked in yellow on the attached paper.

  1. There are other published papers that report the use of 450 nm blue laser and a gigabits per second data rates. For example, the 2017 paper entitled "blue laser diode enables underwater communication at 12.4 Gbps" by T-C Wu et al. provides experimental result to demonstrate > 10 m transmission. It is not clear if these papers are not cited because they do not provide path loss and sensitivity figures.

Response: This paper focuses on the study of long-range wireless optical communication technology underwater, i.e. the system needs to have a high link loss or a low received optical power. The paper entitled Blue laser diode enables underwater communication at 12.4 Gbps" by T-C Wu et al. does not provide a specific link loss or water quality parameters used in the experiments. The main focus of this paper is on underwater high-rate signal transmission, which is slightly different from the focus of our paper. So we did not cite the 2017 paper entitled "Blue laser diode enables underwater communication at 12.4 Gbps" by T-C Wu et al.

  1. The connection between the detailed analysis included in this paper and the experimental results is not clear. The text should clearly show how the analysis is used or verified.

Response: To show the connection between the detailed analysis included in this paper and the experimental results, we compare the calculated results of the BER equation (22) with the experimental data in Figure 15.

  1. Discussions of results could be improved, with proper justifications and conclusions. A marked-up manuscript is attached that points out a few of these locations. The paper should be properly edited to improve consistency and reduce grammatical errors. The attached manuscript highlights many places where obvious grammatical corrections are warranted.

Response: We have revised the grammatical errors as follows:

Line 47, revise ‘date delay’ to ‘time delays’.

Line 68, revise ‘going’ to ‘is’.

Table 1, revise ‘1mJ at 1.5KHz’ to ‘1.5W’.

Line 86, delete ‘the’.

Line 94: revise “measured various water types’ optical characteristics parameters’’ to ‘The optical characteristics parameters of various water types’.

Line 95, revise ‘Table 1’ to ‘Table 2’.

Line 109, revise ‘in’ to ‘and show in’.

Line 115, revise ‘Where’ to ‘where’.

Line 121, revise ‘Table 1’ to ‘Table 2’.

Line 126, revise ‘a scintillation effect and phase change’ to ‘optical scintillation effect and phase change’.

Line 129, revise ‘the influence of optical scintillation’ to ‘optical scintillation effect’.

Line 132, revise ‘proven’ to ‘proved’.

Line 139, revise ‘proven’ to ‘shown’.

Line 151, revise ‘is silent sate ’ to ‘is in a silent state’.

Line 155, revise ‘Ts ’ to ‘Ts’.

Line 164, revise ‘carries out ’ to ‘carried out’.

Line 220, revise ‘Fig. 3 ’ to ‘Figure 3’.

Line 255, revise ‘detailed values ’ to ‘value’.

Line 257, delete ‘given’.

Line 271, revise ‘Where ’ to ‘where’.

Line 276, revise ‘extremely different’ to ‘extremely difficult’.

Line 277, delete ‘as a way’.

Line 287, revise ‘Fig. 6’ to ‘Figure 6’.

Line 307, revise ‘less 384.62MHz’ to ‘less than 384.62MHz’.

Line 308, revise ‘less 37.88m’ to ‘less than 37.88m’.

Line 318, revise ‘2min’ to ‘2 minutes’.

Line 331, revise ‘the electrical pulse of finite width and a fixed amplitude is output’ to ‘the output electrical pulse has a finite width and a fixed amplitude’.

Line 364, revise ‘To output pulse and dead time’ to ‘The output pulse and dead time’.

Line 367, revise ‘due to’ to ‘because’.

Line 381, revise to ‘an ideal Poisson process whose mean and variance are always equal’.

Line 444, revise ‘bellows’ to ‘is below’.

Line 446, This conclusion can be get from Figure 14.

Line 467, revise ‘under’ to ‘at’.
